# Oral Palatability Testing of a Medium-Chain Triglyceride Oil Supplement (MCT) in a Cohort of Healthy Dogs in a Non-Clinical Setting

**DOI:** 10.3390/ani12131639

**Published:** 2022-06-26

**Authors:** Benjamin Andreas Berk, Rowena Mary-Anne Packer, Julia Fritz, Holger Andreas Volk

**Affiliations:** 1Department of Clinical Science and Services, The Royal Veterinary College, Hawkshead Lane, North Mymms, Hatfield, Hertfordshire AL9 7TA, UK; rpacker@rvc.ac.uk (R.M.-A.P.); holger.volk@tiho-hannover.de (H.A.V.); 2BrainCheck.Pet®, Tierärztliche Praxis für Epilepsie, Sachsenstraße, 68167 Mannheim, Germany; 3Napfcheck®, Veterinary Specialist Practice for Small Animal Nutrition, 81377 Munich, Germany; jfritz@napfcheck.de; 4Department of Small Animal Medicine and Surgery, University of Veterinary Medicine, 30559 Hannover, Germany

**Keywords:** MCT, ketogenic, supplement, palatability, nutrition, canine, epilepsy, neurodietetics, tolerance

## Abstract

**Simple Summary:**

The palatability of functional foodsis significant l in therapy compliancewhen used as an adjunct in the management of health conditions such as epilepsy or dementia. The aim of this study was to evaluate the palatability and tolerance of an MCT oil as a dietary supplement in healthy dogs compared to a tasteless, purified control oil. An at-home single-bowl palatability test with three phases lasting five days eachto the usual base-diet, was conducted. The data were collected from nineteen healthy dogs. There was no difference found in the average food intake or intake-ratio between food with and without oil supplementation or between the two oil supplementation groups. In conclusion, MCT oil given as a short-term supplement is not only well tolerated but palatable in a healthy canine population, with only some changes to eating behaviour.

**Abstract:**

The oral palatability of functional foods such as medium-chain triglycerides (MCT) play a crucial role in owner and patient compliance when used as an adjunct in the management of health conditions such as epilepsy. Despite the promising benefits, the palatability of MCT has not undergone a more recent evaluation in dogs. The aim of this study was to assess the palatability and tolerance of short-term, daily supplementation of a 10% metabolic energy based MCT oil volume compared to a tasteless control oil in healthy dogs. An at-home, randomized, double-blinded, controlled single-bowl palatability test with three five-days phases was conducted. Data were collected from nineteen healthy dogs via study visits, feeding diary and eating questionnaires. No difference in the average food intake or intake ratio between food with and without oil supplementation or between the two oil groups was found. The mean food intake time was longer under MCT. In conclusion, MCT oil given as a short-term supplement is well tolerated and palatable in a healthy canine population, with only some changes in eating behaviour. Our results support earlier evidence that MCT oil is a well-tolerated additive in the nutritional management of different diseases such as epilepsy or dementia in dogs.

## 1. Introduction

Palatability is defined as the physical and chemical properties of a diet that are associated with promoting or suppressing feeding behaviour during the pre-absorptive or immediate post absorptive phase [1]. Taste, odour, and texture are the principal components that govern the palatability of a food substance [2]. Measuring palatability in animals can be challenging due to the subjective nature of the animal’s response to food, and the inability to receive direct self-reported feedback, as is often relied upon in human studies. Different standardised methods for testing palatability and food preferences exists in companion animals [3,4,5]. Nevertheless, palatability testing is typically performed in one of two ways: (1) the single-bowl, also known as an ’acceptance’ test, or (2) the two-bowl forced choice method, known as a ’preference’ test [5]. Acceptance is defined as the consumption of a sufficient number of calories to sustain body weight and performance, whilst preference is a graded choice for one item over another and infers a comparison between the two items [5].

Palatability testing is most often carried out on pet food to ensure that companion animals accept the food they are offered in order to sustain a healthy body weight. However, it is equally important in pharmaceutical and nutraceutical agents as the level of consumption has a major impact on the convenience of administration, which in turn influences both patient and owner compliance. This is of particular importance for chronically administered agents, e.g., in chronic disease management. Medium chain triglyceride (MCT) oil is one such nutraceutical agent with a long-standing history of efficacy in several different physiological aberrations in both animal, such as during lactation in sows, or in the cognitive dysfunction of dogs [6,7,8,9,10], and human studies, such as epilepsy, allergy, and gastrointestinal disorders [11,12,13]. MCT is a structurally modified lipid that is predominantly composed of medium chain fatty acids (MCFA) with a carbon chain length ranging from C8 to C12 [13]. To date, a large body of research has demonstrated that the ingestion of MCT is beneficial to the alleviation of numerous health disorders ranging from cognitive impairment, epilepsy, colitis, and muscular dystrophy, as well as exhibiting anti-tumour effects and an overall reduction in insulin resistance, inflammation, and overall body fat accumulation in different species [8,14,15,16,17].

To date, palatability studies of MCT in the canine population are lacking, with only a handful of studies published in small animal nutrition. While some trials showed no effect on palatability in rats [18], a significant aversion to MCT was found in both chickens [19] and cats [9]. Palatability in dogs has only been reported to a limited extent or as an additional secondary observation under laboratory conditions, but not related to the main study objectives in an irregulated setting at home. Matulka and colleagues [20] explored MCT palatability as a secondary factor at 5 to 15% dietary MCT content in a dry food when exploring MCT toxicity. Dogs receiving 15% MCT in feed composition had lower levels of food intake due to suspected palatability issues. Previously, Beynen et al. [7] reported no palatability issues or food refusal in dogs at 11% MCT content in a kibble diet when evaluating the plasma lipid concentrations, macronutrient digestibility, and mineral absorption effects of MCT. Conversely, Pan et al. reported a poor acceptance of MCT at 9% in a kibble diet when investigating the efficacy of MCT on cognitive dysfunction syndrome [16]. The inconsistency in these observations may be due to several different factors ranging from the type of diet (wet vs. dry) and the composition of the diet, but also the equipment (bowls, etc.) MCT has been offered. Depending on the base diet and/or material of the bowl used to offer MCT, oxidative changes of MCT can occur and lead to potential altered taste and odour. The main limitation is the fact that the reported observations of the palatability evaluations were often made in studies where the primary research question was not centred around palatability.

Similarly, tolerability was mostly not mentioned or only roughly discussed as a secondary outcome in those studies above. As tolerability refers to the degree to which the overt adverse effects of a drug or food ingredient can be tolerated by a patient, it is another important part in the process of application and dosing recommendations for a common dietary supplement. In cats, an MCT diet enriched with caprylic acid (C8) led to eating refusal that resulted in extensive weight loss within the first three days [9]. In dogs, no adverse events have been reported within the above levels of dietary content. The pancreatic response using MCT oil in a fluid form added to a standardized manufactured kibble diet did not affect pancreatic parameters in blood [21].

In summary, none of these studies explored the palatability or tolerability effects of an MCT oil given in a fluid oil form as a daily dietary supplement to a variable baseline diet in dogs. Despite one study [21], all the studies have been conducted with MCT as an ingredient in a manufactured kibble diet, but none of these studies explicitly investigated palatability and/or tolerability in dogs.

Therefrom, the purpose of this study was to examine the effects of MCT oil supplementation given in a fluid form on palatability and tolerability in a healthy canine population, consisting of a sample made up of a variety of breeds and ages, in a non-clinical environment.

## 2. Materials and Methods

### 2.1. Study Population

Dogs were recruited through various social media platforms as well as study posters displayed at BrainCheck.Pet^®^, a veterinary practice for epilepsy in small animals in Germany. Only companion dogs were recruited, and only one dog per household could enroll in the study. An online pre-study questionnaire (hosted on SurveyMonkey^®,,^ Dublin, Irland) was completed for each dog, capturing all relevant data pertaining to clinical history and the dog’s recent health status. A minimum age of one year was required to participate. The body condition score was taken and was required to be between 4 to 6 [22]. Dogs were excluded from the recruitment process if they had any chronic or acute renal, hepatic, gastrointestinal (including dental problems), respiratory, or cardiac disease; an acute or surgical condition at the time of enrolment; or for bitches, known as or suspected to be pregnant or lactating. Dietary history was also collected before enrollment. Dogs needed to be MCT naïve and were not recruited when MCT was already a component (in any kind: oil, kibble diet, or similar) of their usual diet.

Dogs with a clinically unremarkable physical and neurological examination were enrolled. Dogs were then randomly assigned to either MCT or control dietary supplement (DS) group. A neurological exam was deemed necessary, as the processing of the complex taste experience involves the careful coordination of multiple brain regions, whereby palatability is reflected within distinct epochs of cortical and amygdaloid responses. Permuted block randomisation was employed using Randomization.com (http://www.randomization.com (accessed on the 6 November 2016)). A unique study case number, ascending in a chronological order of enrolment, was assigned, and used for identification purposes.

### 2.2. Dietary Supplements

Both the control and test oils were commercially available and suitable for human consumption. The control oil was extra-virgin tasteless olive oil with 8 kcal per ml containing 11% saturated, 11% polyunsaturated, and 78% monounsaturated fat as stated in the nutritional declaration label (Filippo Berio, Italy, Batch No LE194M04). The test oil was an MCT-DS purified from palm and rapeseed oil, and consisted of 50–65% caprylic acid (C8) and 30–50% capric acid (C10) with 8.37 kcal per ml from 93% saturated fatty acids (End GmbH, Germany, Batch No. L 16 M12).

### 2.3. Oil Supplementation

The quantity of DS oil given daily was calculated by first establishing the Resting Energy Requirement (RER) using the metabolic body weight in kilograms: $$\mathrm{RER}=70\times {\left(\mathrm{Body}\;\mathrm{Weight}\;\mathrm{in}\;\mathrm{kg}\right)}^{\frac{3}{4}}$$

RER was then multiplied by a coefficient based on the life stage and neuter status to calculate the Daily Energy Requirement (DER):e.g., DER of normal neutered adult=1.6×RER

The quantity of oil was calculated by dividing the DER by the number of kcal per millilitre of each oil type and the final volume represented 10% isoenergetic requirement per day:$$\mathrm{Oil}\;\mathrm{amount}\;\mathrm{at}\;10\%\;\mathrm{isoenergic}\;\mathrm{requirement}\;\mathrm{per}\;\mathrm{day}=\left(\frac{\mathrm{DER}}{\mathrm{kcal}\;\mathrm{per}\;\mathrm{ml}\;\mathrm{of}\;\mathrm{oil}}\right)\times 0.1$$

Previous studies have demonstrated that the addition of MCT oil at 5.5% content intake exhibits antiepileptic properties and a significant reduction in seizure frequency in canines [15] or canine dysfunction [16,17]. It was therefore deemed of critical importance to evaluate palatability of MCT oil at clinically relevant concentrations. Calculated volumes of both oils were divided into two equal portions and decanted into brown, sealed syringes ensuring blinding of all parties involved. The DS allocation (MCT-DS vs. control-DS) was randomized and only available to the study nurse, who was also the dietary supplement dispenser. Dog owners, investigators, other nurses, and statisticians involved were blinded throughout the study. The total required number of syringes for the trial was then dispensed to the owners with detailed instructions for the supplementation protocol. Owners were instructed to give the oil twice daily to their dog’s usual baseline diet during the intervention period, and to ensure the oil was homogenously mixed into the food to ensure a uniform distribution. The owners were also instructed to keep the baseline diet consistent throughout the whole study’s duration and that concomitant changes or additional dietary supplementation would lead to exclusion from the study. Dogs were fed in isolation in households with multiple dogs to remove the influence of competitive feeding.

### 2.4. Experimental Design

A single-bowl palatability test system, with three periods of five days each, was chosen to evaluate the daily supplementation of one commercially available MCT oil for palatability in terms of acceptance and tolerance in comparison to a control. The study involved a fifteen day, short-term, prospective, randomized, double-blinded, and controlled dietary trial comparing the chosen MCT-DS to a standardized control oil DS. Dogs enrolled in this clinical trial were initially fed for five days with their usual diet composition (phase 1, day 1–5), followed by another five days in which they were supplemented with either MCT-DS or control-DS alongside their normal diet (phase 2, day 6–10). The final study phase was another five days of normal feeding without any supplementation (phase 3, day 11–15) to record any aftereffects and changes in eating behaviour (Figure 1).

On enrolment visit (day 0) and at the end of each dietary intervention period (day 5, day 10, day 15), the veterinarian conducted a physical and neurological examination as well as a recording of the body mass of each dog.

### 2.5. Assessment of Eating Behaviour and Palatability

A combination of a standardized feeding diary (Appendix A) and established owner-based questionnaire was used to assess palatability and tolerance per dietary intervention phase. Over the entire study period of 15 days, owners were asked to record the amount of baseline diet food that was fed to dogs (in grams), the time taken to consume the food (seconds), and any side-effects (Figure 1).

Each owner received detailed instructions about how to measure different variables in this study. The amount of food fed was recorded by weighing the food bowl before and after feeding with a regular digital kitchen gram scale and a standardized ceramic bowl. Feeding time was recorded from the start of feeding to the end of feeding with a stopwatch. Each meal was video recorded from start to end and evaluated by the researchers. The time span that was statistically evaluated included the time spent on inspecting the food and nasal insufflation. Feed weight was documented in grams and feeding time in seconds. During phases two and three, the owners were instructed to pay particular attention to changes in behaviour such as grass consumption, water intake, food intake, and overall changes in appetite, as well as gastrointestinal changes including the softening of faecal matter and changes in the frequency of urination and defecation. Acute adverse events including vomiting, diarrhea, abdominal pain, flatulence, and eructation were also recorded at the end of each study phase. The owners were advised to record the data immediately after feeding or witnessing behavioural changes to ensure data accuracy.

Based on this data, the following variables were calculated per dog and compared by phase and group:

Average food intake (aFI) in grams;

Average food intake time (aFIT) in seconds;

Food intake ratio (FIR1/2 = [Food Phase X/(Food Phase X + Food Phase X + 1)]) per phase;

Mean food intake ratio (mFIR) [23].

A FIR of 0.5 represents equivalent consumption of food with each oil. As a result, an FIR greater than 0.55 during the oil supplementation phase was defined as an increase in food consumption, whereas an FIR of less than 0.45 was identified as a decrease in food intake in comparison to phases 1 or 3. All parameters together allow for an appropriate global evaluation of the effects of MCT oil as DS on palatability. Owner-based questionnaire data complemented these objective calculations. Any abnormal events occurring after enrolment were reported to the investigator via the questionnaire and documented as a potential adverse reaction (Table 1). Concomitant treatments administered during the study were also recorded.

### 2.6. Statistical Analysis

Comparisons between study variables (aFI, aFIT, FIR, and mFIR) for the MCT-DS and control-DS groups were conducted using a paired Student’s t-tests for normally distributed data, and Mann–Whitney U-Test or Wilcoxon matched-pairs signed rank test for data that did not follow a normal distribution (GraphPad Prism^®^; STATCON GmbH; Schulstr. 2; Witzenhausen, Germany). Normal distribution of the data was assessed using the Kolmogorov–Smirnov normality test. The association between two (normally distributed) continuous variables was analysed using Pearson’s correlation coefficient analysis. Continuous variables following a normal distribution were reported as means ± SEM. Whereas non-normally distributed continuous variables were expressed as box and whisker plots with the mean depicted as a central line, the box representing interquartile range (IQ25–75), and whiskers showing the range of the data. All comparisons were two-sided and *p* < 0.05 was considered statistically significant.

## 3. Results

### 3.1. Demographics

In total, 19 of the 20 recruited dogs completed the three phased trial. One dog was removed two days after study initiation due to the owner’s non-compliance. The following breeds were included: Australian Shephard, Chihuahua, Border Collie, Dwarf Schnauzer, Flat-Coated Retriever, German Shepherd, Golden Retriever, Labrador, Pug, Shetland Sheepdog, and crossbreeds (*n* = 6). The study population consisted of 7 males, of which 4 were neutered and 3 intact, and 12 females, of which 3 were neutered and 9 intact. The dogs had a mean age of 7.8 (±4.3) years and a mean weight of 18.6 (± 10.9) kg. Almost all the dogs (89.2%, n = 17) were vaccinated on a regular basis and at least dewormed once/year (n = 18). No chronic diseases were reported for all study dogs. Table 1 of the supporting information summarises the baseline characteristics of all the recruited dogs as well as the medications in use during the study’s duration (Table 2).

Dogs were fed twice daily (68.4%) with a commercially available dry food diet (94.7%) purchased through an online vendor. A small portion of the dogs irregularly received wet food (26.3%) in addition to their dry feed. Table 3 of the supporting information provides an overview of the study population’s dietary routine, treat provisioning, and any existing supplementation use (Table 3).

### 3.2. Palatability

Due to the dropout of one dog at the beginning of the trial, the MCT-DS was tested on ten dogs and the control-DS on nine dogs. While seventeen dogs showed no acceptance problems subjectively, two dogs in the control-DS initially refused to eat the feed with an oil homogenously mixed into the feed. Based on the owner’s report, both went back to normal feeding on the same day (day 6 of phase 2) without further problems. No further acceptance issues were noted throughout the study.

The average food intake (%) did not significantly differ between the baseline food intake at phase one and the oil supplementation at phase two, or the supplementation at phase two and the after-effect at phase three (MCT-DS: *p* = 0.52, control-DS: *p* = 0.93). When the oil was supplemented in phase two, there was no difference in the aFI between the two DS types (MCT-DS: 85.53 % (±10.49), control-DS: 88.89 % (±11.11), *p* = 0.65). However, it was noted that dog 14 presented with palatability problems and a general lack of acceptance of MCT-DS. The percentage of food consumed by dog 14 rapidly declined on the first day of phase two to 40% and gradually increased back to 100% on the first day of phase three (Figure 2a).

The mFIR between phases one and two showed a small increase between the control (mFIR1: 0.5) and MCT group (mFIR1: 0.51) (Figure 2b). Similarly, the mFIR between phases two and three showed a very small decrease between the control (mFIR2: 0.5) and MCT group (mFIR2: 0.49). There was no difference in the mFIR between phases one and three (mFIR3: 0.5). The overall food intake ratio did not differ significantly between MCT-DS and control-DS compared to the reference (MCT-DS: 0.5 vs. control-DS: 0.5, *p* = 0.56) or after-effect phase three (MCT-DS: 0.5 vs. control-DS: 0.5, *p* = 0.65). Overall, the small difference between the groups was not statistically significant, indicating that there was no food preference between the three phases. There were no changes in body weight within the DS-group (*p* = 0.66) or the entire study population (*p* = 0.78) between the start and completion of the study (15-day period).

### 3.3. Eating Behaviour

Our results showed that the addition of an oil into a dog’s feed, irrespective of the type of oil, resulted in a significant increase in the average food intake time. The aFIT in phase one where only the baseline diet was used was 124.72 s (±161.18), whereas the addition of an oil in phase two resulted in an increase in the aFIT to 197.21 s (±242.18). A comparison of the aFIT between phase two and phase three, where the oil was once again omitted, showed a decrease in the aFIT from 197.21 s (±242.18) to 173.41 s (±63.45), indicating that the exclusion of oil from the feed resulted in a decrease in the aFIT (Figure 3a). A two-tailed Wilcoxon matched-pairs signed rank test showed that the aFIT between phase one and phase two was statistically significant (*p* = 0.008), as well as the difference between phases two and three (*p* = 0.041). Comparing the aFIT between MCT oil and the control olive oil in phase two only showed a statistical increase in the aFIT (*p* = 0.038). The aFIT increased from 144.31 (±174.14) seconds in the control-DS group to 244.8 (±291.6) seconds in the MCT-DS group (Figure 3b).

### 3.4. Gastrointestinal Tolerance

Overall, there were no reports of adverse events such as vomiting, diarrhea, or abdominal pain associated with the consumption of either DS, indicating that both oils were well tolerated. All the dogs were clinically unremarkable at all study visits and showed no signs of acute adverse events. However, dogs that received MCT-DS exhibited grass consumption (40%, n = 4, *p* = 0.71) and flatulence (30%, n = 3, *p* = 0.66) more frequently than those receiving the control-DS, while those receiving the control-DS exhibited softened faeces (44%, n = 4, *p* = 0.71) or an increased appetite (33%, n = 3, *p* = 0.71) more frequently than those receiving MCT-DS. These symptoms disappeared completely in phase 3 in MCT-DS group; however, those in the control-DS group persisted. The occurrence of an adverse event was not statistically associated with the type of DS (X^2^ = 8.067; df = 12, *p* = 0.78).

## 4. Discussion

Our results demonstrated that there were no significant adverse side effects associated with the consumption of MCT oil at 10% isoenergetic requirement divided into two portions per day over the course of five days. This is consistent with previous toxicity studies of MCT oil in canines wherein no adverse effects were reported while using MCT oil supplementation with percentages of up to 15% over the course of 91 days [20]. This may be due to the ingestion of MCT oil within a mixed meal, since the simultaneous consumption of MCT oil with carbohydrates has been shown to abolish adverse effects [24]. Nonetheless, caution should still be exercised with the long-term use of MCT oil as detrimental effects on bone health have been associated with the use of the caprylic acid component of MCT oil [25]. Thereby, MCT should be potentially avoided in growing animals and investigated in the future during different life stages. Although this risk was postulated in rats, and has not been replicated yet in dogs, MCT application should be properly monitored in animal nutrition.

The disappearance of physiological effects, such as flatulence and the increased grass eating habits in the after-effect phase of the MCT-DS group, could be due to MCT’s rapid metabolism and blood clearance rate. Triglycerides are hydrolysed to release fatty acids, and the rate of metabolism of free fatty acids is very much dependent upon the carbon chain length. Ge et al. [26] demonstrated that the maximal reactive rate of lipoprotein lipase hydrolysis on MCT was 2.2 times faster than on long chain triglycerides (LCT), and the reactive rate for the hydrolysis of hepatic triglycerides lipase on MCT was 1.2 times faster compared to LCTs. Given that MCFA have a shorter chain length, they are delivered directly to the liver via the hepatic portal vein to undergo the β-oxidation process [13]. This indicates that MCTs are hydrolysed faster and more completely and are therefore cleared from the circulation at a faster rate [13]. However, the reason why dogs eat grass has not been clarified yet; palatability issues should be considered in some patients. One common assumption is that dogs eat grass to relieve upset stomachs [27]. Although no significant frequency of adverse events was found in this study, the baseline diet with its specific fibre content might play a relevant role in the therapeutic approach of MCT supplementation and should be therefrom considered in small animal nutrition.

Conversely, in the control-DS group, physiological changes such as softened faeces persisted well into the after-effect phase. Olive oil, selected as the study control, is known to be predominantly composed of long chain fatty acids (LCFA), namely palmitic (C16), oleic, and linoleic (C18) fatty acids [11]. Due to the increased carbon chain length, olive oil not only takes longer to undergo hydrolysis but also takes a completely different metabolic pathway. Unlike MCFA, LCFA are absorbed by the lymphatic system and transported as chylomicrons into the systemic circulation where they can be incorporated into adipose tissue and muscle [13]. Given that LCFA enter the circulation, they are more likely to produce a long-term physiological effect compared to MCFA.

Previously, MCT oil has been found to have an effect on satiety whereby shorter chain triacylglycerols such as that of MCT are more satiating than long chain triacylglycerols [28]. However, our results did not show any statistically significant difference in appetite between dogs in the MCT-DS or control-DS groups. This may be due to a number of different factors. The satiety effects of MCT oil have recently been shown to be dependent on whether MCT oil is consumed in a solid or liquid form, with greater satiating effects exerted when consumed in liquid form [10]. The majority of the dogs in this study were fed a solid dry food diet, which may explain the reduced effects on satiety; however, it may also be possible that the satiating effects of MCT may present after a prolonged period of exposure. Future work could therefore aim to establish the exact feeding requirements of MCT oil, including the impact of heating the oil during the preparation on the overall physiological effects on the target organism. Furthermore, establishing the exact feeding parameters needed to exert a satiating effect would be of great benefit for dogs that require strict weight management.

A food substance with an over 90% voluntary acceptance rate in the target species is deemed as palatable [4]. The absence of significant difference in the average food intake and food intake ratio between the DS groups or the entire study population demonstrates that MCT oil has passed the acceptability test in healthy canines. Although our results showed that the dogs didn’t appear to exhibit a preference towards food with or without the addition of an oil supplement, there was an obvious difference in the eating behaviour. A comparison of the aFIT in the feeding phases with and without the oil showed that the time to complete consumption increased when the oil was added, indicating that the general addition of the oil, irrespective of the type of oil, resulted in an increased feeding time. This is likely to be predominantly caused by the taste-modifying effects of fats upon flavour release mediated through: (1) changes in the partitioning of the flavour compounds among the food, saliva, and the taste receptors; (2) physical interference with diffusion processes affecting the tastant’s access or binding to the taste receptors; or (3) changes in the rate of regeneration of the interfacial surfaces required for the release of sapid compounds into the surrounding media [29]. Alternatively, this can simply be a novelty effect that would disappear after time with ongoing habituation to the new feed composition [4]. The supplementation of MCT oil was well accepted by both study groups; therefore, it can be deemed mostly palatable.

Our findings indicate that on average, dogs took the longest to eat food supplemented with MCT oil when compared to olive oil. These findings are consistent with the observations made by Lynch and colleagues who showed that coconut oil, similar in composition to MCT oil, was considerably more potent in suppressing taste perception compared to sunflower oil [29]. This is speculated to be due to the more viscous consistency of coconut oil at a given oral temperature compared to sunflower oil. However, previous reports have shown that fats are well known to modify the taste balance by altering the perception of sweetness, saltiness, sourness, and bitterness, with the extent of the modification being strongly dependent not only on the type of oil used but also the testing species involved [9]. Therefore, the type of baseline diet used will have a strong influence on taste perception and should be considered in future investigations.

There are several limitations associated with this study. The study design did not use a controlled baseline diet, and instead implemented the existing diets of the study dogs, including any prior supplementation that the dogs were originally fed. Given that the alteration of taste perception can depend on the testing species, the food composition can be further refined to ensure optimal acceptance rates without changes to eating behavior. Although our choice of using the owner selected baseline diet may be seen as a limitation, it may also be an advantage, as introducing a completely new diet to the dog may induce neophobia and may result in skewed results where the aversion may be to the new diet instead of the added supplement [4]. In contrast, neophilia could also have impacted our results. As an adaptive trait for domestic dogs towards man, novel-food might be chosen with a higher preference [3,30]. However, given that the food intake ratio between both oils did not differ significantly, a neophiliac effect might be very minimal. Another limitation is the short duration of the investigation. Although we leaned on previousstudies in the duration of investigation [3,23], long-term studies are needed to evaluate oil supplementation, when gradually introduced, and/or how eating behavior might change over time.

In this study, only one specific type of MCT oil containing a mixture of only C8 and C10 fatty acids was tested. Previous reports have shown that C10 β-oxidation is markedly lower than that of C8 in neuronal cells, and this reduced oxidation rate of C10 may lead to its accumulation in the brain providing anti-seizure effects. Furthermore, MCT oil’s chain length has been shown to influence serum lipid parameters [13]. It is therefore imperative that a greater understanding of the impact of the different medium chain lengths on the downstream physiological effects is developed, as this will be crucial in helping veterinarians to plan the most appropriate dietary regimens. Moreover, this study only involved a small number of healthy dogs, with testing for a short duration. Therefore, it would be useful to recruit a larger sample capturing a greater diversity of breeds and ages, and to use dogs with chronic gastrointestinal diseases or other conditions known to affect taste perception, to ensure that the results are widely applicable and translatable. Based on our results, we would recommend veterinarians to introduce MCT oil as DS only when owners are instructed to keep their usual baseline diet stable. Further changes in their dog’s diet can be made depending on the nutritional approach.

Despite the limitations of this study, the results demonstrate that MCT oil is not only palatable to dogs in a non-clinical environment but that it does not result in any significant adverse effects during short term use. This supports its more frequent use in small animal nutrition. As MCT oil has been shown to be a useful adjunct in the nutritional management of a number of different conditions in dogs, including epilepsy [14,31] and cognitive dysfunction [17,32], MCT is still a promising nutritive component and should be further explored for its effects on other diseases in companion animals.

## 5. Conclusions and Recommendations

The palatability of health-promoting nutraceuticals has a substantial impact not only on the convenience of administration and compliance from both the owner and the patient, but also on the success of the nutritional approach. This is particularly important for chronically administered nutraceuticals over long periods of time. Our results have demonstrated the good palatability of MCT oil administered at 10% isoenergetic requirement per day in a canine population. In addition, MCT oil was shown to be well tolerated without any substantial adverse events. This presents a promising opportunity to introduce MCT oil as an additional complementary nutritive treatment option for dogs with chronic disorders including idiopathic epilepsy and cognitive dysfunction, where the benefits of MCTs have been demonstrated.

## Figures and Tables

**Figure 1 animals-12-01639-f001:**
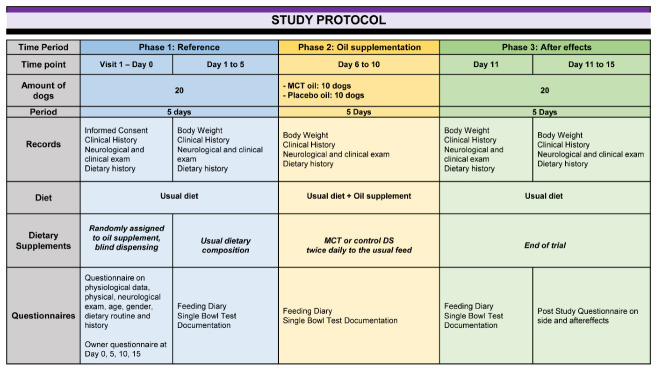
Study protocol: A single-bowl palatability test system. Three periods of five days each were chosen to evaluate the daily supplementation of one commercially available MCT oil for palatability in terms of acceptance and tolerance in comparison to a control oil. Phase 1 constituted to the reference point where no oil was supplemented, phase 2 included the integration of the oil in the feed, and phase 3 was used to observe any long-term effects of the oil post integration. The oil used in the test was randomized and blinded to all participants except the dispensing study nurse. Relevant data to assess MCT oil as a suitable dietary supplement was collected at four study visits.

**Figure 2 animals-12-01639-f002:**
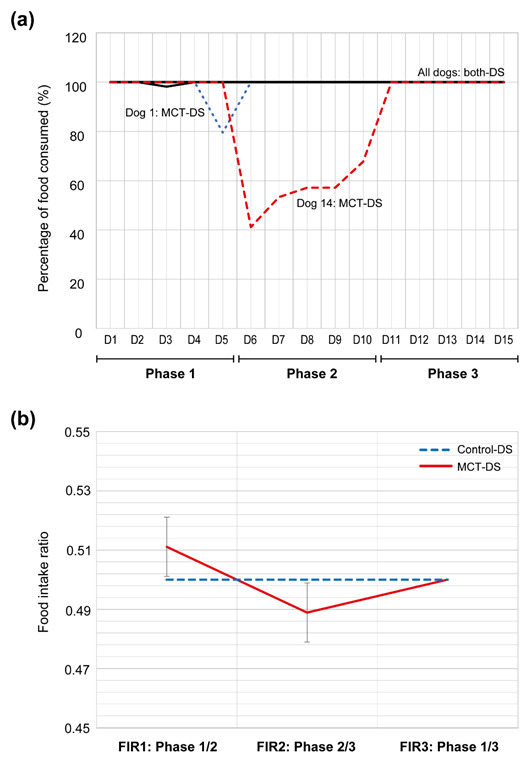
Food consumption plots. (**a**) Plot showing the percentage of food consumed in each study phase. (**b**) Plot showing the food intake ratio per phase (FIR). A calculated FIR of 0.50 represents an equivalent consumption rate. An FIR greater or lower than 0.50 indicates an increase or decrease in the quantity of food consumed, respectively. The FIR of MCT-DS did not exhibit a statistically significant difference when compared to the control-DS throughout all the study phases.

**Figure 3 animals-12-01639-f003:**
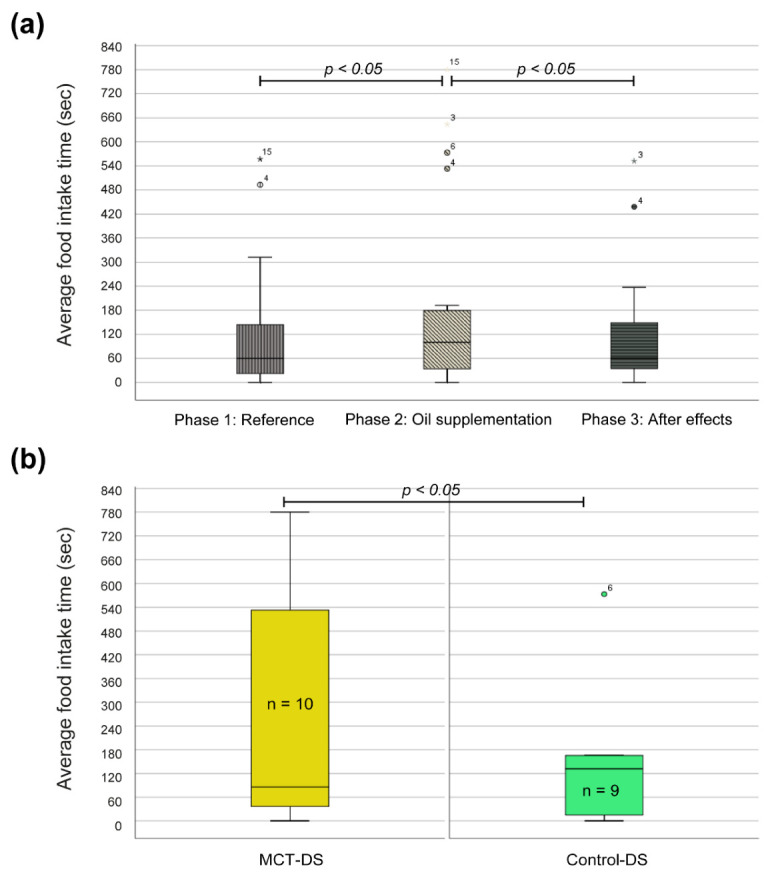
Effects of oil supplementation on the average food intake time (aFIT). (**a**) Box and whisker plots of the aFIT per study phase measured in seconds. (**b**) Box and whisker plots of the aFIT differentiated by oil type. The central lines of the box and whisker plots represent the median, while the lower and upper limits of the box represent the 25th and 75th percentiles, respectively, and whiskers represent the minimum and maximum. Two-tailed Wilcoxon tests were conducted to compare aFIT per phase and independent t-test for comparison of both DS types.

**Table 1 animals-12-01639-t001:** Adverse events reporting.

No.	Question	Not Present	Slightly Present	Moderately Present	Severely Present
1	Did the dog experience any vomiting?			X	X
2	Did the dog experience any diarrhea?			X	X
3	Did the dog have soft faeces?			X	X
4	Did the dog experience problems with swallowing?				X
5	Was grass eating behavior present?	X		X	X
6	Did the dog experience any abdominal pain?			X	X
7	Was there an increase in water intake?			X	X
8	Was there an increase in urination?	X		X	X
9	Was there an increase in defecation?			X	X
10	Did the dog experience flatulence?	X		X	X
11	Did the dog experience eructation?			X	X
12	Was there an increase in food intake?	X		X	X
13	Was there an increase in appetite?	X		X	X
14	Was there a decrease in appetite?			X	X

**Table 2 animals-12-01639-t002:** Baseline characteristics of the study population *.

Dog	Study Arm	Dog Breed	Age (Years)	Weight (kg)	Sex (F/M)	Neuter Status (Y/N)	Medication
1	**MCT**	Boxer–Pointer Mix	15	27	F	Y	-
3		Pug	3	12	M	N	-
4		Sheltie–Jack Russel Mix	14	14	F	Y	-
7		Labrador	2	14	F	N	-
10		Dwarf Schnauzer	10	8	F	Y	-
13		Flat-Coated Retriever	3	32	M	N	-
14		Labradoodle	1	30	M	N	-
15		Labrador	6	23	F	Y	-
16		Sheltie–Jack Russel Mix	14	12	F	Y	Caniphedrine
18		Parson Russel Terrier Mix	7	6	F	Y	Omega 3 fatty acids
2	**Control**	German-Shepard–Husky Mix	8	28	F	Y	-
5		Shetland Sheepdog	3	9	F	Y	-
6		German Shephard	10	42	M	Y	Cimicoxib
8		Labrador Mix	9	10	F	Y	-
9		Border Collie	7	23	M	Y	-
11		Chihuahua	5	2	F	N	-
17		Golden Retriever	13	30	M	Y	-
19		Pug	9	9	M	N	Antiacida
20		Australian Shephard	9	22	F	N	-

* Dog number 12 was excluded from the study due to owner non-compliance.

**Table 3 animals-12-01639-t003:** Dietary routine and treat provisioning in the study population *.

Dog	Study Arm	Baseline Diet Composition	Baseline Diet Source	Dietary Routine	Treats	Prior Supplementation	Prior Supplementation Routine
1	**MCT**	Dry food	Online vendor	Twice per day	Daily	Fatty acids, oils	Monthly
3		Raw diet	Pet store	Twice per day	Daily	Never	-
4		Dry food Wet food (tinned)	Pet store	Once per day	Daily	Other	Daily
7		Dry food	Veterinary practice	Twice per day	Daily	Never	-
10		Dry food	Online vendor	Twice per day	Never	Other	Weekly
13		Dry food	Online vendor	Twice per day	Daily	Never	-
14		Dry food	Online vendor	Thrice per day	Weekly	Fatty acids, oils	Weekly
15		Dry food Wet food (tinned)	Pet store	Thrice per day	Daily	Never	-
16		Dry food Wet food (tinned)	Pet store	Once per day	Daily	Other	Daily
18		Dry food	Veterinary practice	Twice per day	Monthly	Fatty acids, oils	Daily
2	**Control**	Dry food	Online vendor	Twice per day	Daily	Fatty acids, oils	Monthly
5		Dry food	Pet store	Twice per day	Daily	Never	-
6		Dry food	Online vendor	Twice per day	Daily	Fatty acids, oils	Daily
8		Dry food	Supermarket	Ad libitum	Daily	Other	Monthly
9		Dry food	Pet store	Twice per day	Daily	Never	-
11		Dry food	Veterinary practice	Twice per day	Monthly	Never	-
17		Dry food Wet food (tinned)	Online vendor	Twice per day	Daily	Other	Daily
19		Dry food Self-cooked food	Pet store	Thrice per day	Monthly	Never	-
20		Dry food	Online vendor	Twice per day	Weekly	Never	-

* Dog number 12 was excluded from the study due to owner non-compliance.

## Data Availability

Data available on request due to restrictions eg privacy or ethical. The data presented in this study are available on request from the corresponding author. The data are not publicly available due to funding body reasons.

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
