# Peer review of "Oral Palatability Testing of a Medium-Chain Triglyceride Oil Supplement (MCT) in a Cohort of Healthy Dogs in a Non-Clinical Setting"

_animals, 2022, doi:10.3390/ani12131639_

Round 1

Reviewer 1 Report

Dear authors, 

Very nice job with this manuscript. Below are some suggested edits and some comments for your consideration.

Comment 1: Lines 3 & 4 Is “Oral palatability of medium-chain triglyceride oil in healthy dogs” necessary? The title appears to be already complete without out it. Perhaps consider adding “Oral” at the beginning if this should be included in the title 

Comment 2: Line 40 – Unnecessary space before the period (i.e., ... (Hyde and Witherly, 1993) . 

Comment 3: Line 56  - triglycerides doesn’t need the “s” here 

Comment 4: Line 69 – sentence seems to end abruptly – consider something like “...with only a handful or studies conducted/published” 

Comment 5: Line 69 – show or showed should be used instead of showing 

Comment 6: Line 72 – extent should be used instead of extend 

Comment 6: Line 73 – missing the word “a” before secondary 

Comment 7: Line 74 – 5 to 15% MCT content – should you say dietary MCT content? 

Comment 8: Line 80 - It would be interesting to know whether the way MCT was added to the diet was different - e.g. for some studies was it formulated in as part of the diet and for others was it added as a supplement onto the diet? 

Also, did these studies implement any type of transition to the diets or were the dogs switched from the base diet to the diet with the MCT supplement abruptly? 

Comment 9: Line 82: “...but also the equipment (bowls, etc.) used to offer MCT.”? 

Comment 10: Line 82 – Unnecessary space in “and/ or” 

Comment 11: Line 84 – Unnecessary comma after “fact” 

Comment 12: Line 85 – Unnecessary comma after “studies” 

Comment 13: Line 88 - You explain very well palatability in the introduction but do not explain tolerability. It may be useful to explain this term as well so we understand why that is part of your objective. 

Comment 14: Line 94 – comma unnecessary after “small animals” 

Comment 15: Line 95 when you use the phrase companion animals I think cats and dogs but I think you mean companion dogs (i.e. not working dogs?) You should probably clarify that you only used dogs. 

Comment 16: Line 98 – health status instead of state 

Comment 17: Line 98 Did you have a minimum age requirement for the dogs, or could puppies have been included (lowest age is 1 year old on what is currently labelled table 1) 

Comment 18: Line 100 can remove “, were” so that it reads ”... or for bitches known or suspected to be pregnant or lactating.” 

Comment 15: Line 122 – Did all dogs have an ideal BCS or were there some overweight/obese dogs? 

Comment 16: Line 133 did you mean cognitive dysfunction? 

Comment 17: Line 142 Did you collect a diet history before enrolment to assess current diet and ensure it would be compatible with the additional MCT supplement? For example, did their diet already contain MCT? Were they fed other supplements that would interfere with the taste? Were any of the dogs used to have a diet change frequently so might be more likely to accept a novel supplement? 

Comment 18: Line 145 would instead of will 

Comment 19: Figure 1 – In reference to “Phase 1: Reference” heading – would baseline be a better word for this phase? 

Comment 20: Figure 1 – In reference to “Phase 3: After effects” heading – Post supplementation, instead of after effects? 

Comment 21: Line 170 just wondering if BCS was also assessed 

Comment 22: Line 174 – “during each” instead of “per” 

Comment 23: Line 175 – unnecessary “the”, i.e., “...study period of 15 days, owners were...” 

Comment 24: Line 176 – in grams instead of per grams 

Comment 25: Line 177 add such as and list some examples of side effects 

Comment 26: Line 178 “Owners received detailed instructions about...” 

Comment 27: Line 180 kitchen scale – digital kitchen gram scale? 

Comment 28: Line 181 add “a” in front of stopwatch 

Comment 29: Line 181 Each meal instead of any meal 

Comment 30: Line 184 grams instead of gram 

Comment 31: Line 209 should this be table 1 because it is the first table to appear? Other table 

Comment 32: Line 219 – was instead of were 

Comment 33: Line 236 was dental disease also ruled out? It was not mentioned above in exclusion criteria 

Comment 34: Line 243 Majority of dogs – the can be removed 

Comment 35: Table 2 - For the dogs that were fed three times per day or ad lib, were the owners instructed to feed them only twice per day for the study period? I wonder if this could impact feeding time - if a dog was still fed 3 times and was given the oil in 2 meals. those 2 meals would be a smaller volume. Also on this same thought, dogs fed a homemade diet or raw diet or canned diet would have a larger volume of food to eat since these diets have a higher moisture content. Was the gram weight of the food considered in the eating time? 

Comment 36: Line 282 “...did not differ overall...” or “The overall food intake ration did not differ between...” might work better here 

Comment 37: Line 284 if changing the descriptors for the phases to baseline and post supplement in the table  (i.e., comments 19 & 20), text in the body of the report will need to be updated as well. Phase 1 is already referred to as baseline, so this already makes sense to change in the table 

Comment 38: Line 286 There were no changes... significant not necessary here 

Comment 39: Line 291 resulted in an increase – significant not necessary here 

Comment 40: Line 316 DS instead of DSs 

Comment 41: Line 340 could instead of can 

Comment 42: Line 344 2.2x or 2.2 times, currently redundant 

Comment 43: Line 351 - Owners did not have to record this behaviour during phase 1, correct? Would it have been a limitation to not have a baseline of this behaviour? Perhaps some of these dogs were prone to eating grass already? Was access to grass the same for all dogs? 

Comment 44: Line 352 Although works better than despite 

Comment 45: Line 376 including the impact of heating oil... 

Comment 46: Line 377 – unnecessary comma after preparation 

Comment 47: Line 378 You could also measure satiety hormones as a way to investigate the satiety effect of MCT 

Comment 48: Line 380 - Wondering if all dogs ate their entire allotment of each meal? I don't think I saw this data reported - just the time to eat the meal - but did they all finish their meals? Were owners given guidance on how much base diet to feed or were they to feed according to their usual amounts? 

Comment 49: Line 381 Could also use “The absence of a significant difference...” 

Comment 50: Line 383 “was observed” can be deleted 

Comment 51: Line 396 - Could some additional limitations or areas for further work include 1. a study that investigates palatability but introduces the oil gradually for a few days up to the full dose compared to dogs that start at the full dose? and 2. a longer duration to see if over time the eating time improves - 5 days is quite short 

Comment 52: Line 413 - I think there is a word missing before indicates - this sentence is not clear 

Comment 53: Line 416 - You could also include that in a clinical setting where this supplement would be recommended by the veterinarian the dog would continue to eat its usual diet, and so this is more reflective of a real scenario 

Comment 54: Line 423 “only” can be removed 

Comment 55: Line 442 - perhaps diseases which affect companion animals instead of in Veterinary Medicine? 

Comment 56: Line 459 – B.A.B was the recipient 

Comment 57: general comment - inconsistent formatting with abbreviations (i.e., use of periods) – e.g., line 457 (H.A.V.), line 459 (B.A.B) and line 461 (BAB) 

Comment 58: Lines 462 – patient instead of patients, could also rewrite as “..., carried out the main practical work, recruited patients and...” 

Comment 60: References please make sure you are consistent with formatting.

Author Response

Thank you very much for reviewing our article. We appreciate your feedback and comments. We hope being able to rule out your concerns by delivering more details out study and changed the manuscript as suggested.

  • Comment 1: Lines 3 & 4 – Is “Oral palatability of medium-chain triglyceride oil in healthy dogs” necessary? The title appears to be already complete without out it. Perhaps consider adding “Oral” at the beginning if this should be included in the title

We have changed the title.

  • Comment 2: Line 40 – Unnecessary space before the period (i.e., ... (Hyde and Witherly, 1993).

We have checked the manuscript and adapted where applicable.

  • Comment 3: Line 56  - triglycerides doesn’t need the “s” here

Deleted.

  • Comment 4: Line 69 – sentence seems to end abruptly – consider something like “...with only a handful or studies conducted/published”

Extended.

  • Comment 5: Line 69 – show or showed should be used instead of showing

Changed.

  • Comment 6: Line 72 – extent should be used instead of extend

Changed.

  • Comment 6: Line 73 – missing the word “a” before secondary

Added.

  • Comment 7: Line 74 – 5 to 15% MCT content – should you say dietary MCT content?

Added.

  • Comment 8: Line 80 - It would be interesting to know whether the way MCT was added to the diet was different - e.g. for some studies was it formulated in as part of the diet and for others was it added as a supplement onto the diet?

All studies explored MCT only as a component of a complete kibble diet manufactured by for example Purina or Hills, but did not investigate a dietary oil supplement added to a variable baseline diet. We have made substantial adaptions (line 71 -91) expressing and highlighting more this goal in our study.

  • Also, did these studies implement any type of transition to the diets or were the dogs switched from the base diet to the diet with the MCT supplement abruptly?

In all trials, dogs were directly switched to the new kibble diet with MCT.

  • Comment 9: Line 82: “...but also the equipment (bowls, etc.) used to offer MCT.”?

Changed.

  • Comment 10: Line 82 – Unnecessary space in “and/ or”

Changed.

  • Comment 11: Line 84 – Unnecessary comma after “fact”

Deleted

  • Comment 12: Line 85 – Unnecessary comma after “studies”

Deleted

  • Comment 13: Line 88 - You explain very well palatability in the introduction but do not explain tolerability. It may be useful to explain this term as well so we understand why that is part of your objective.

Thank you, we have added another section and brought both aspects together (Line 87 – 100)

  • Comment 14: Line 94 – comma unnecessary after “small animals”

  • Comment 15: Line 95 – when you use the phrase companion animals I think cats and dogs but I think you mean companion dogs (i.e. not working dogs?) You should probably clarify that you only used dogs.

Changed.

  • Comment 16: Line 98 – health status instead of state

Changed

  • Comment 17: Line 98 – Did you have a minimum age requirement for the dogs, or could puppies have been included (lowest age is 1 year old on what is currently labelled table 1)

Due to reports about the effects on bone formation in rats, we decided one year as minimum age requirement. Further informtion added (line 113)

  • Comment 18: Line 100 – can remove “, were” so that it reads ”... or for bitches known or suspected to be pregnant or lactating.”

Deleted.

  • Comment 15: Line 122 – Did all dogs have an ideal BCS or were there some overweight/obese dogs?

BCS was collected and had to be between 4 and 6. Further details has been added to the manuscript (Line 114) based on Laflamme.

  • Comment 16: Line 133 – did you mean cognitive dysfunction?

Ypun are right. For example, such a disease as CCD, should have been excluded, when not already diagnosed. CCD can significantly influence such results.

  • Comment 17: Line 142 – Did you collect a diet history before enrolment to assess current diet and ensure it would be compatible with the additional MCT supplement? For example, did their diet already contain MCT? Were they fed other supplements that would interfere with the taste? Were any of the dogs used to have a diet change frequently so might be more likely to accept a novel supplement?

Dietary history was also collected before enrolement and dogs were not recruited, when MCTs were already a component (in any kind: oil, kibble diet or similar) of their diet. We defined hem as MCT-naive. Of course, we agree that other supplements might interfere with MCT oil supplement; hwoever as we did not find significant differences overall or cases being out of the mass in statistics, we do not expect here a massive impact. Additional information were added to the manuscript (Line 118-120).

  • Comment 18: Line 145 – would instead of will

Changed.

  • Comment 19: Figure 1 – In reference to “Phase 1: Reference” heading – would baseline be a better word for this phase?

We decided to use reference instead of baseline, as this should highlight to which „reference“ we relate to in our calculations. Furthermore we did not to interfere with the terminus „baseline diet“.

  • Comment 20: Figure 1 – In reference to “Phase 3: After effects” heading – Post supplementation, instead of after effects?

You are right. However, as „post supplementation“ might indirectly highlight, that observed post events are „logically“ connected to the type of supplement, we decided to avoid this terminus in our study. In this case, we planned to label and discuss as neutral and objective as possible for owners. Due tot hat, we kept this terminology in our manuscript.

  • Comment 21: Line 170 – just wondering if BCS was also assessed

BCS was assessed, but due to the range defined for study inclusion we have not listed it.

  • Comment 22: Line 174 – “during each” instead of “per”

Changed.

  • Comment 23: Line 175 – unnecessary “the”, i.e., “...study period of 15 days, owners were...”

Changed.

  • Comment 24: Line 176 – in grams instead of per grams

Changed.

  • Comment 25: Line 177 – add such as and list some examples of side effects

These are already listed in line 202 to 208.

  • Comment 26: Line 178 – “Owners received detailed instructions about...”

Changed.

  • Comment 27: Line 180 – kitchen scale – digital kitchen gram scale?

  • Comment 28: Line 181 – add “a” in front of stopwatch

  • Comment 29: Line 181 – Each meal instead of any meal

  • Comment 30: Line 184 – grams instead of gram

  • Comment 31: Line 209 – should this be table 1 because it is the first table to appear?

  • Comment 32: Line 219 – was instead of were

Comment 27 to 32 accordingly changed.

  • Comment 33: Line 236 – was dental disease also ruled out? It was not mentioned above in exclusion criteria

This was included into gastrointenstinal disorders, but is now separately mentioned in brackets (line 116).

  • Comment 34: Line 243 – Majority of dogs – the can be removed

Changed.

  • Comment 35: Table 2 - For the dogs that were fed three times per day or ad lib, were the owners instructed to feed them only twice per day for the study period? I wonder if this could impact feeding time - if a dog was still fed 3 times and was given the oil in 2 meals. those 2 meals would be a smaller volume. Also on this same thought, dogs fed a homemade diet or raw diet or canned diet would have a larger volume of food to eat since these diets have a higher moisture content. Was the gram weight of the food considered in the eating time?

Thanks for this comments. The owners were instructed to keep the dietary regimes (amount, composition, type) as implemented before the study. For better comparison, the oil was only added to the breakfast and dinner meal, volume was not changed and the composition was the same without. Only in one case (Dog 8) we changed from ad libitum to twice daily about two weeks before recruitment. On statistics, the dog appeared not to differ from the study population.

In discussion with the relevant owners at the scheduled appointments, the eating behaviour appeared not significantly different between the meals within a day. Comparing the variables we have measured between dogs with two and those with three meals a day, no statistically significance was found beeing different or impacting our results (aFIT, p=0.65; mFIR = 0.71).

  • Comment 36: Line 282 – “...did not differ overall...” or “The overall food intake ration did not differ between...” might work better here

Changed.

  • Comment 37: Line 284 – if changing the descriptors for the phases to baseline and post supplement in the table (i.e., comments 19 & 20), text in the body of the report will need to be updated as well. Phase 1 is already referred to as baseline, so this already makes sense to change in the table 

Thank again for this suggestion. Please see comment 19 in regards to explanation keeping those termini.

  • Comment 38: Line 286 – There were no changes... significant not necessary here

  • Comment 39: Line 291 – resulted in an increase – significant not necessary here

  • Comment 40: Line 316 – DS instead of DSs

  • Comment 41: Line 340 – could instead of can

  • Comment 42: Line 344 – 2x or 2.2 times, currently redundant

Comments 38 to 42 - Changes have been made.

  • Comment 43: Line 351 - Owners did not have to record this behaviour during phase 1, correct? Would it have been a limitation to not have a baseline of this behaviour? Perhaps some of these dogs were prone to eating grass already? Was access to grass the same for all dogs?

Thanks for this comment. We have collected data about this beahviour on initial recuitment. None of the dogs showed grass eating before, otherwise we would have not recruited this dog due tot he potential of beeing associated with gastrointestinal issues. All dogs were routinely walked and had access to grass.

  • Comment 44: Line 352 – Although works better than despite

  • Comment 45: Line 376 – including the impact of heating oil...

  • Comment 46: Line 377 – unnecessary comma after preparation

Comments 44 – 46 Changes have been made.

  • Comment 47: Line 378 – You could also measure satiety hormones as a way to investigate the satiety effect of MCT

Thank you very much! This was another idea we had. But we planned doing this without any invasive techniques and further ethical approvmend, thus we skipped this idea. But I agree, this should be another goal for future studies.

  • Comment 48: Line 380 - Wondering if all dogs ate their entire allotment of each meal? I don't think I saw this data reported - just the time to eat the meal - but did they all finish their meals? Were owners given guidance on how much base diet to feed or were they to feed according to their usual amounts?

Thanks for your question. In order to calculate the food intake ratio (FIR), the amount the dog had eaten in grams per phase was necessary and set it into relation giving an assessment about palatability. The amount in grams was used to calculate the ratio per phase between each day (phase 1 – Day 1 vs. phase 2 Day-1) and then as average value for the entire period per phase again. In all calculations, no differences were seen, leading tot he conclusion that the food intake was stable and the palatability was not influenced. They fed the usual amounts with the prescribed oil amount.

  • Comment 49: Line 381 – Could also use “The absence of a significant difference...”

  • Comment 50: Line 383 – “was observed” can be deleted

Comment 49, 50: Changes have been made (line 399 to 400)

  • Comment 51: Line 396 - Could some additional limitations or areas for further work include 1. a study that investigates palatability but introduces the oil gradually for a few days up to the full dose compared to dogs that start at the full dose? and 2. a longer duration to see if over time the eating time improves - 5 days is quite short

Thanks for this comment. We agree and added this to the manuscript. (Line 439 – 442)

  • Comment 52: Line 413 - I think there is a word missing before indicates - this sentence is not clear

  • Comment 53: Line 416 - You could also include that in a clinical setting where this supplement would be recommended by the veterinarian the dog would continue to eat its usual diet, and so this is more reflective of a real scenario

  • Comment 54: Line 423 – “only” can be removed

  • Comment 55: Line 442 - perhaps diseases which affect companion animals instead of in Veterinary Medicine?

  • Comment 56: Line 459 – B.A.B was the recipient

  • Comment 57: general comment - inconsistent formatting with abbreviations (i.e., use of periods) – e.g., line 457 (H.A.V.), line 459 (B.A.B) and line 461 (BAB)

  • Comment 58: Lines 462 – patient instead of patients, could also rewrite as “..., carried out the main practical work, recruited patients and...”

Comments 52 to 58 have been addressed. Recommendation as in comment 53 added (line 454 – 457)

  • Comment 60: References – please make sure you are consistent with formatting.

References are updated and adapted with the MDPI Chicago style.

Reviewer 2 Report

This study presents a palatability trial of a MCT supplement using client-owned dogs of varying ages and breeds eating a variety of baseline diets. This is a novel approach, as opposed to using laboratory dogs and/or a population of dogs all fed the same base diet. 

 Given the heterogeneity of the population an their diets, I would have liked to see a larger sample size - were sample size calculations performed ad hoc?

Oil supplementation: the dose was established as providing 10% of DER requirement based on calculation of RER, but how did this translate to the energy intake of the dogs? Were the owners instructed to feed to meet DER or did they continue feeding according to how they were feeding prior to the study? I would expect a few dogs would be consuming less than DER, and some more, which would alter the proportion of energy intake from the oils - thus some dogs were potentially slightly under- or over-dosed, which could impact palatability based on the tolerance of each dog.

I am interested as to why time spent inspecting food and sniffing food was not included in the analysis. These behaviours represent key features of palatability and food acceptance. It would be interesting to see a comparison of time spent inspecting and sniffing food between the control and MCT supplementation.

A few specific comments and corrections are in the attached edited manuscript.

Author Response

Comment 1:  Given the heterogeneity of the population an their diets, I would have liked to see a larger sample size - were sample size calculations performed ad hoc?

Thank you very much for reviewing our article. We appreciate your comments and hope being able to rule out your concerns by delivering more details about sample and sample size calculation

We agree that the study population and the baseline diet was overall very heterogeneous. However, as we aimed to gain a most realistic picture how an usual, commercial MCT oil might influence given as a daily dietary supplement (DS) the acceptance of their normal baseline diet, or how it may change their eating behaviors, we especially tried to recruit a very diverse study population with diverse types of baseline diet. We consider the fact that we have still found some significant effects on the eating behavior, but none on eating preference, as realistic and representative. Nevertheless, we still agree that a bigger sample size should be aimed in future trials with standardized baseline diet to get further knowledge about how MCT DS influences the specific type of baseline diet and how this may impact our study variables. Some study variables might depend on the type of baseline diet.

The sample sizes calculation was performed as combination of two different methods ad hoc.

First of all, it based on the number of individuals significant findings under dietary intervention in similar study design were detected (Law et al. 2015: N=21). This was supported by another study, which similarly suggested twenty-two dogs in each group being sufficient for showing significant differences between two diet groups, in this case for for seizure frequency (Patterson et al. 2005: N=22). Secondary, we have used the program G*Power 3.1 (University of Dusseldorf, Germany) to double check for sample size calculation for comparing two means from two paired samples using Wilxocon signed-ranked test (N=37). In addition, we have used for the same statistical test PASS (NCSS) (N= 32, with 30% dropout (N=10)) Please find attached the calculation we have done.

However, in regard to the known problems of calculating power and sample size for our aimed statistical test (Gwowen et al. 2007) and the recommended 3R’s in clinical trials with animals, we have focused on the sample sizes from representative data and evidence-based studies. Therefrom, we have started with 20 individuals and decided to extend the study population depending on the statistically significant findings we made. In this case, we got significant findings in eating preference and aspects of eating behavior already with 19 dogs.

In summary, we agree that the small sample size should be considered as limitation of this study and need to be re-evaluated in future clinical trials about the palatability of MCT oil in small animal nutrition.

Comment 2:  The dose was established as providing 10% of DER requirement based on calculation of RER, but how did this translate to the energy intake of the dogs? Were the owners instructed to feed to meet DER or did they continue feeding according to how they were feeding prior to the study?

Thanks for your comment. Unfortunately, despite aspects of safety consumption (Hall et al. 2012, James ert al. 2009, Matulka et al 2009), there are truly not many recommendations about the proper dose of MCT oil supplement in dogs with specific health conditions. Therefrom, we orientated the ME based amount on the previous study from Pan et al. 2012, Law et al. 2015, Pan et al. 2018 and palatability recommendation from different other studies, but also to meet nutritional guidelines established by the Association of American Feed Control Officials (AAFCO) and the National Research Council (NRC).

For calculation of the daily energy requirement, we have used the recommendation by NRC (2006). Owners were instructed to feed their dog stable as prior to the study and add the calculated oil amount to their dog’s feed. We agree that this can impact the palatability based on the tolerance of each dog in some cases, however we did not expect in this investigation a significant influence due to the short-term design. Aside this aspect, we have not documented any food refusion or partly eaten meals during the entire study periods. Although, we cannot exclude that in some case dogs were over- or underdosed, we do not expect relevant influences in a healthy population of dogs. Based on previous research, initial palatability issues have been reported initially at a level of 15% in healthy beagles in a lab environment (Matulka et al. 2009).   

To avoid over- or underdosing in clinical practice, it may be better to start from a lower calculated MER and add as needed to maintain optimal body weight; or calculate the energy intake the dog is getting now keeping its body weight stable and using it for further dietary intervention. Both methods are implemented small animal nutrition in clinical practice.

Comment 3: I am interested as to why time spent inspecting food and sniffing food was not included in the analysis. These behaviours represent key features of palatability and food acceptance. It would be interesting to see a comparison of time spent inspecting and sniffing food between the control and MCT supplementation.

Thanks for your idea and feedback. Owners were asked to document any behavior in deviation from normal eating behavior with their normal diet in the provided feeding diary. Based on our documentation and final evaluation, no significant behavior was observed. However, as eating time increased, we agree that there might have occurred some behavioral traits (as sniffing and food inspection) owner cannot interpret in comparison to dogs in a lab environment. Nevertheless, such changes have also not been reported in safety or toxicity studies from the past. Based on that we considered this increase in eating time (might include those behavioural changes) more into neophobia or changes in the oral textures. But still, this needs more exploration in future research, especially in long-term studies.

Comments in the manuscript: Line XX + Changes

·         Line 58: Further details have been added.

·         Line 66: These findings were made in different species, especially lab animals, dogs and humans.

·         Line 183: Thanks for (very much) for highlighting this sentence. We had changed it as performed: The time span statistically evaluated did not include the time spent on inspecting the food and nasal insufflation.

·         Line 336: Manuscript has been screened for octanoic and decanoic acid. Adaption to constant terminology has been made.

·         Line 338: Sentence about translation of effects on growth has been added.

·         Line 350: Corrected.
